# Revealing new pathways for the reaction of Criegee intermediate CH₂OO with SO₂
Cangtao Yin ✉ & Gábor Czakó ✉

Criegee intermediates play an important role in the tropospheric oxidation models through their reactions with atmospheric trace chemicals. We develop a global full-dimensional potential energy surface for the $CH_2OO + SO_2$ system and reveal how the reaction happens step by step by quasi-classical trajectory simulations. A new pathway forming the main products ($CH_2O + SO_3$) and a new product channel ($CO_2 + H_2 + SO_2$) are predicted in our simulations. The new pathway appears at collision energies greater than 10 kcal/mol whose behavior demonstrates a typical barrier-controlled reaction. This threshold is also consistent with the ab initio transition state barrier height. For the minor products, a loose complex $OCH_2O \cdots SO_2$ is formed first, and then in most cases it soon turns into $HCOOH + SO_2$, in a few cases it decomposes into $CO_2 + H_2 + SO_2$ which is a new product channel, and rarely it remains as $\cdot OCH_2O \cdot + SO_2$.

Understanding the chemistry of the Criegee intermediates ($R_1R_2COO$) has become one of central topics in atmospheric research recently. Criegee intermediate reactions directly lead to many end products important to atmospheric chemistry, such as hydroxyl radicals, organic acids, hydroperoxides, and aerosols[1–11]. Increasing evidence shows that the reaction between Criegee intermediates and sulfur dioxide makes significant contributions to atmospheric $H_2SO_4$ and thus to aerosol formation[12–18]. Many advanced experimental techniques such as laser flash photolysis (LFP), photoionisation mass spectrometry (PIMS), ultraviolet (UV) absorption, laser induced fluorescence (LIF), and cavity ring down spectroscopy (CRDS) are employed to suggest that $CH_2O + SO_3$ are the main products of the $CH_2OO + SO_2$ reaction[19–28]. Theoretical studies also imply that $CH_2O + SO_3$ are the dominant products by forming a five-member-ring heteroozonide complex, $CH_2OOS(O)O$[29–34]. Rice−Ramsperger−Kassel −Marcus (RRKM)/master equation (ME) simulations predicted that at least 97% yield is $CH_2O + SO_3$ at 295 K[32]. Although the main atmospheric removal mechanism for $CH_2OO$ is the reaction with the water dimer, the negative temperature dependence for the $CH_2OO + SO_2$ reaction in conjunction with the decrease in water dimer concentration at lower temperatures means that $CH_2OO$ can play an enhanced role in $SO_2$ oxidation in the atmosphere at lower temperatures[28]. In addition, some Criegee intermediates can survive in humid condition, for example, methacrolein oxide, an isoprene-derived Criegee intermediate[16].

Apart from the main products, theoretical studies provided evidence that the minor products $HCOOH + SO_2$ appear by means of rearranging and decomposing of $CH_2OOS(O)O$[32,35]. In this process, the $SO_2$ molecule catalyzes isomerization of $CH_2OO$ to $HCOOH$. An early experimental

study[36] isotopically double labeled $CH_2{}^{18}O^{18}O$ and reacted it with unlabeled $SO_2$. They observed the singly labeled product $HCO^{18}OH$, which means that the $SO_2$ molecule participates in the reaction process and exchanges one oxygen atom with $CH_2OO$. However, in a recent work, $HCOOH$ was unobserved with transient infrared absorption spectroscopy and its branching ratio was estimated to be < 5%[37].

Vereecken et al.[31] predicted theoretically that the $CH_2OO + SO_2$ reaction could also produce $\cdot OCH_2O \cdot$, which is a singlet bisoxy radical, an open form of dioxirane. The $SO_2$ molecule also serve as a catalyst during this process. However, the barrier from $\cdot OCH_2O \cdot$ to $HCOOH$ is only about 3 kcal/mol[38]. Besides, energetically speaking, $\cdot OCH_2O \cdot$ is very hard to survive since $HCOOH$ is about 95 kcal/mol lower[32,38]. In addition, Kuwata et al.[32] confirmed that its branching ratio is less than 1% by RRKM/ME simulations.

In the present work, we seek a global full-dimensional potential energy surface (PES) for the $CH_2OO + SO_2$ reaction system with high-level computational precision and explore possible new channels and products by means of quasi-classical trajectory (QCT) simulations. It is believed that the full-dimensional PES greatly promotes comprehensive understanding of the reaction mechanism.

First, we select some representative geometries to test several quantum computational methods and efficient and accurate methods that provide reliable energies are found. Second, we develop a global full-dimensional PES of this system according to high-level couple-cluster ab initio method with the help of ROBOSURFER program package[39]. Third, we perform QCT simulations on the PES and the dynamic properties we are interested in can be statistically obtained from the computed trajectories. At the end we present our results and make a discussion.

MTA-SZTE Lendület Computational Reaction Dynamics Research Group, Interdisciplinary Excellence Centre and Department of Physical Chemistry and Materials Science, Institute of Chemistry, University of Szeged, Rerrich Béla tér 1, Szeged, Hungary. ✉e-mail: cangtaoyin@foxmail.com; gczako@chem.u-szeged.hu

**Article**

## Methods

### Quantum computation methods

When it comes to the quantum chemistry computational theories, MP2[40] is our first choice for a system with around ten atoms due to its fast calculation of thousands of energies needed for the global PES. However, the accuracy of MP2 is not satisfactory in nowadays. Therefore, coupled-cluster theory should be applied to obtain energies that are more precise. As for the basis set, aug-cc-pVDZ (aVDZ)[41] is commonly used in the MP2 theory. There can be plenty of different combinations for the coupled-cluster theory and basis set. In this article, three of them are tested and listed in Table 1. All the quantum chemistry calculations in this study are performed with the MOLPRO program package[42].

Before doing the calculation, what we have to consider is which geometries should be chosen to test and find the best computation level. Besides the reactants and products, the five-member-ring heteroozonide complex $CH_2OOS(O)O$ that every reaction channel passes, is also important for the $CH_2OO + SO_2$ reaction. There are two conformers in this complex since the direction of the extra oxygen atom can be either *endo* or *exo* respect to the COOSO ring.

In Table 1, it is presented that the higher quantum calculation levels are, the more precise the energies are. As expected, the MP2/aVDZ cannot give a chemically accurate result but it is super-fast thus it could be our initial choice for the development of global rough PES. As for the coupled-cluster theory, a large difference occurs between CCSD-F12b/cc-pVDZ-F12 and CCSD(T)-F12b/cc-pVDZ-F12, while a small one happens between CCSD(T)-F12b/cc-pVDZ-F12 and CCSD(T)-F12b/cc-pVTZ-F12 due to the fast basis-set convergence of the explicitly-correlated F12b method. Therefore, CCSD(T)-F12b/cc-pVDZ-F12 should be chosen to be our final calculation level to guarantee the accuracy and efficiency, which takes around ten minutes for each ab initio calculation at the level of CCSD(T)-F12b/cc-pVDZ-F12.

At the CCSD-F12b/cc-pVDZ-F12 level of theory, the $T_1$ diagnostic values are 0.0178 and 0.0183 for complex *endo* and complex *exo*, respectively, whereas the corresponding $T_1$ values are 0.0179 and 0.0184 with the larger cc-pVTZ-F12 basis set, in order.

Single reference methods are safe for all the exit channels since the $T_1$ values of the products are 0.015 ($CH_2O$), 0.016 (HCOOH), and 0.027 ($OCH_2O$), respectively. The one with more multi-reference character is the reactant $CH_2OO$, whose $T_1$ value is 0.044 (Vereecken et al.[31]) or 0.042 (our calculation), which is right on the dividing line between whether one must use multireference methods or not. Although some of the stationary points have higher $T_1$ values (e.g., TS-12a in Kuwata et al.[32]), the most important geometries, i.e., the five-member-ring heteroozonide complex *endo* and *exo*, have $T_1$ values below 0.02, which are low enough to be treated with a single-reference method. In addition, as Yin and Takahashi[43] investigated, ignoring the static correlation only causes an underestimation of the TS energy by about 1 kcal/mol in the unimolecular reaction of $CH_2OO$. Vereecken et al.[44] and Anglada et al.[45] also confirmed that some other Criegee related systems can be treated with single-reference methods. As a conclusion, although Criegee reactions have some multireference character at certain regions of the configuration space, in practice the use of single-reference methods is sufficient for this system.

### Approach to develop the global PES

Our approach to develop PES is in general divided into three steps. (1) A rough global PES, whose energy values have a large error, is obtained using the fast method MP2/aVDZ in the ROBOSURFER program package. (2) CCSD(T)-F12b/cc-pVDZ-F12 is used to recalculate the energies of the geometries in step 1. Although it takes a longer time, the accuracy of energies is quite satisfying and guaranteed. Nevertheless, it is expected that there will be more failure points than at the MP2 level, which could cause some "holes" (unphysically deep minima) on the obtained PES in the area of our interest. (3) In order to fill the possible holes in the PES, the ROBOSURFER program package is employed again. It picks the geometries along the trajectory before it fails, i.e., goes into a hole. Then the program calculates the energy, evaluates its usefulness, and then adds it into the current geometry set. After hundreds of iterations, very few trajectories fail and most holes are filled in. The ideal PES can be obtained from the above three steps.

The ROBOSURFER is a useful and easy-to-use program package, whose main steps are as follows: (1) Analytical PES function using Monomial Symmetrization Approach (MSA)[46] is determined in the initial geometry set (see next section). In MSA, the PES function is an expansion of $y_{ij} = \exp(-r_{ij}/a)$, here $r_{ij}$ presents the distances between two arbitrary atoms and $a$ is a parameter that controls the asymptotic behaviour of the PES. Our group have tested it and found out that 2 bohr is usually a good choice for neutral systems, including a similar system $CH_2OO + NH_3$[47]. The $\exp(-r_{ij}/a)$ function is a common choice when fitting the potential energy *vs* distance of two atoms, since its second-order polynomial expansion behaves like Morse potential and it guarantees that the interaction energy goes to zero when two atoms are far away from each other. The weighted least squares is employed with $E_0/(E + E_0)$ as the weighting factor, here $E$ is the potential energy of the respective geometry relative to the global minimum of the geometry set, and $E_0$ is set to be 0.1 hartree. (2) Using the movement trajectories of atoms on the PES to generate some new geometries automatically. (3) Finding the most promising geometries from newly generated ones. (4) Calculating the ab initio potential energies using the MOLPRO program package. (5) Adding the geometries and the corresponding energies to the data set after a comprehensive inspection. (The subprogram of ROBOSURFER, called ADDPOINTS, adds geometries into the fitting set. It tries to reduce the scaled fitting error for all geometries while keeping the fitting set as small as possible. More details can be found in ref. 39.). Afterwards, the new analytical PES function can be determined again and the iteration continues to improve the PES further.

### Initial geometry set

The initial geometry set is generated similarly to our previous work[47–49]. Firstly, we move the Cartesian coordinates of the 31 stationary points[32] of the $CH_2OO + SO_2$ reaction by a random distance of 0–0.4 Å to produce 500 geometries for each stationary point. As for the entrance regions, the distance between $CH_2OO$ and $SO_2$ is randomly set between 3 Å and 8 Å, which

---

**Table 1 | ZPE-corrected energies of the test geometries: major products $CH_2O + SO_3$, minor products $HCOOH + SO_2$, and the five-member-ring heteroozonide complex (*endo* and *exo*), respective to the reactants $CH_2OO + SO_2$, in kcal/mol.[a]**

| Geometries | MP2/aVDZ | CCSD-F12b/cc-pVDZ-F12 | CCSD(T)-F12b/cc-pVDZ-F12 | CCSD(T)-F12b/cc-pVTZ-F12 | Kuwata et al.[32] | ATcT v. 1.130[51] |
|---|---|---|---|---|---|---|
| $CH_2O + SO_3$ | −66.58 | −81.66 | −77.14 | −76.34 | −74.57 | −74.75 |
| $HCOOH + SO_2$ | −123.89 | −120.68 | −116.37 | −116.19 | −115.67 | −115.39 |
| complex *endo* | −39.06 | −38.84 | −35.23 | −35.61 | −35.21 | |
| complex *exo* | −38.45 | −37.73 | −34.28 | −34.72 | −34.32 | |

[a] For the MP2/aVDZ and CCSD-F12b/cc-pVDZ-F12 energies, the geometries are optimized at the stated level of theory; for the CCSD(T)-F12b/cc-pVDZ-F12 and CCSD(T)-F12b/cc-pVTZ-F12 energies, the geometries are optimized at the CCSD-F12b/cc-pVDZ-F12 level.

---

generates 1000 geometries. Same method is applied for the three possible exit regions. At last there are 19,500 geometries in total. In order to get an initial rough PES within a relatively short time, the energies of these geometries are calculated at the MP2/aVDZ level with the MOLPRO program package. Only 187 geometries failed due to HF convergence issues which can be considered negligible. Moreover, we exclude those geometries with too high energies that are way out our interest, i.e., 400 kcal/mol higher than the global minimum, which corresponds to the minor products $HCOOH + SO_2$ according to Table 1. Since some geometries are very distorted with super high energies, they can be harmful to the analytical PES. It is necessary to exclude those geometries so a critical value of energy above the global minimum (product) should be set. In our experience 400 kcal/mol is usually a good choice. At the end, 17,245 geometries and the corresponding energies are retained. The full-dimensional analytical PES function consists of 10220 fitting coefficients in the fifth-order expansion.

## PES development at low and high levels
First we use the MP2/aVDZ level of theory to calculate the ab initio potential energies with MOLPRO and the ROBOSURFER program package to iteratively improve the PES. After 418 iterations we obtain a PES consisting of 38,716 geometries and the corresponding energies.

Then we move on to the CCSD(T)-F12b/cc-pVDZ-F12 level of theory. During the recalculation, 2.7% of the points (around one thousand) failed in the coupled-cluster iterations and 37,676 points are well reserved. To avoid possible holes on the PES, the ROBOSURFER program package is used again. This time 188 iterations are conducted to lower the percentage of the unphysical trajectories to less than 1%. Our final coupled-cluster PES is therefore developed from 46,411 geometries.

## PES evaluation and test
To show the performance of the full-dimensional PES, we adapted the 1-D scan method, i.e., elongating and shortening the C-S bond of an optimized geometry, while keeping the other degrees frozen, to evaluate the accuracy of the PES. The results are shown in Fig. 1. Four different potential energy 1-D scan curves are drawn relative to the reactants $CH_2OO + SO_2$. (1) The C-S bond of the optimized complex *exo* geometry is elongated and shortened, while other degrees of the two parts are frozen. (2) The optimized reactants, $CH_2OO$ and $SO_2$ are separated at the distance of 1 Å to 10 Å, in terms of C-S distance. (3) The optimized major products, $CH_2O$ and $SO_3$ are separated at the distance of 1 Å to 10 Å, in terms of C-S distance. (4) The optimized minor products, HCOOH and $SO_2$ are separated at the distance of 1 Å to 10 Å, in terms of C-S distance.

Four asymptote and well energies from Fig. 1 are listed in Table 2. Comparing the PES values with ab initio ones shows a perfect chemical accuracy for this analytical full-dimensional PES except the asymptote of complex *exo*. Luckily this asymptote is very high in energy (over 60 kcal/mol) and out of our interested region (up to 30 kcal/mol). Table 2 also shows the positions of the minima located at the C-S distance for PES and ab initio. The differences are within 0.1 Å except the minor products case, which is 0.15 Å.

The reactants zero-point energy (ZPE) is also important in our following QCT dynamical study. Our analytical PES gives 19.29 and 4.42 kcal/mol of ZPEs for $CH_2OO$ and $SO_2$, which are very close to the values from ab initio calculations, i.e., 19.67 and 4.49 kcal/mol by CCSD(T)-F12b/cc-pVDZ-F12. Furthermore, 99% of trajectories obtained by QCT simulation (next section) provide physically correct results, which also indicates a good behavior of our new analytical PES from another perspective.

The optimized Cartesian coordinates and energies obtained on the PES are available in Supplementary Data 1.

## QCT dynamic simulations
QCT method has always been considered as a common and useful approach when it comes to studying the kinetic and dynamic properties of chemical reactions. The motion of atoms in reactions controlled by classical mechanics and the inclusion of reactants' ZPE form the QCT method. It is regarded as a computationally easier and faster method than quantum methods. In addition, it also provides us with the visual movement of atoms so that we can get insight into the dynamics of the reaction by analyzing the motion of atoms during collisions.

In order to show our settings of the QCT initial conditions[50] more intuitively, a picture is depicted in Fig. 2. As for the X/Y/Z-direction distance, it represents the distance between the center of mass of the two reactants, i.e., $CH_2OO$ and $SO_2$. The initial condition gives the positions and velocities for each atom, in a way that the energies of the reactants correspond to their ZPE values. The forces are obtained from the PES by computing numerical gradients. Then the positions and velocities of the next step (0.0726 fs) can be calculated using the velocity-Verlet algorithm. Besides ZPE, no other quantum effects are included in the process of simulations. If the distance between the farthest atoms is greater than the largest initial one by 1 bohr, the propagation will be terminated. Two thousand trajectories for each $b$ value will be performed to produce statistically accurate results. The collision energy is the initial X-direction translational energy in the trajectories, i.e., the relative motion energy of the two reactants, $CH_2OO$ and $SO_2$. It is not a temperature because the energy corresponding to a temperature should be distributed among the three directions of translation, three rotations, and $3N - 6$ vibrations, where $N$ is the number of the atoms.

**Fig. 1 | 1D-scan potential energy evaluation.**
Comparisons of potential energies as a function of $d_{C-S}$ (distance between atom C and atom S) obtained from the analytical PES and CCSD(T)-F12b/cc-pVDZ-F12 ab initio calculations. Left panel: entrance-channel 1D scans, right panel: exit-channel 1D scans.

**Table 2 | The ab initio (PES) energies, in kcal/mol, of the asymptotes and the minima of the 1D-scans shown in Fig. 1**

|  | complex *exo* | reactants | Products 1 | Products 2 |
|---|---|---|---|---|
| Asymptote energy | 63.26 (67.04) | 0.00 (0.00) | −77.46 (−77.49) | −118.08 (−117.86) |
| Well energy | −34.25 (−33.48) | −7.72 (−8.84) | −78.72 (−78.28) | −123.87 (−123.58) |
| Well position (Å) | 1.78 (1.78) | 2.53 (2.62) | 3.19 (3.25) | 2.23 (2.38) |

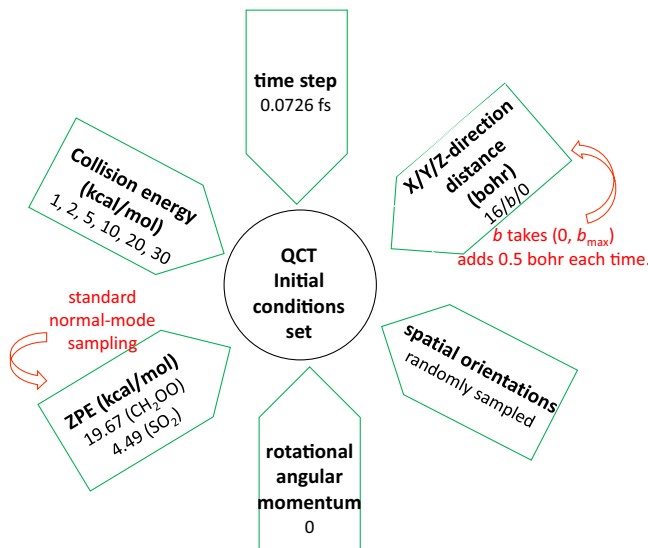

**Fig. 2 | Initial conditions of the QCT simulation.** Six initial conditions of the QCT simulations are presented.

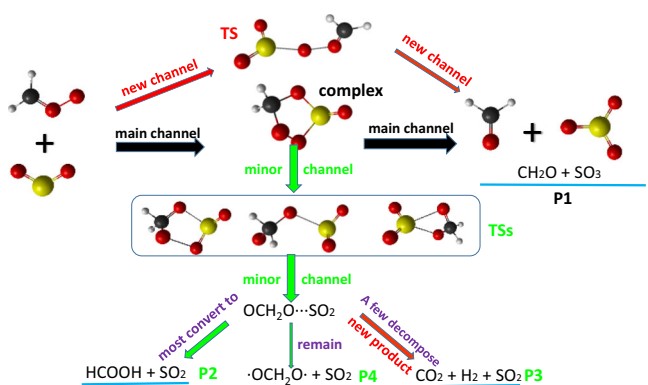

**Fig. 3 | Diagram of the reaction pathways.** Four products in $CH_2OO + SO_2$ reaction named as P1, P2, P3, and P4. For P1, our QCT simulation uncovers a new channel. Furthermore, a new product P3 are obtained during the simulation.

## Results and discussions

Four products named P1 ($CH_2O + SO_3$), P2 ($HCOOH + SO_2$), P3 ($CO_2 + H_2 + SO_2$), and P4 ($\cdot OCH_2O \cdot + SO_2$) for the $CH_2OO + SO_2$ reaction are observed in our QCT simulations. P1 is the main product and the rest are the minors. In addition to the main channel through a five-member-ring heteroozonide complex, a new channel with a chain transition state (TS) forming P1 is predicted (shown in the top of Fig. 3). For the minor products, once the five-member-ring heteroozonide complex passes through one of the three TSs in the rectangular box of Fig. 3, it will first form a loose complex $OCH_2O \cdots SO_2$, but in the most cases it soon turns into P2, in a few cases it decomposes into P3, and rarely it remains as P4. The full reaction pathways are presented in Fig. 3. It is worth pointing out that P3 is a new product in this reaction and has not been studied before. Following the trajectories forming P3, we see that as $OCH_2O$ and $SO_2$ move away from each other, $OCH_2O$ splits into two parts, i.e., $CO_2$ and $H_2$. Finally, the product consists of three parts. We have not found any TS for this mechanism, thus it is probably a barrierless process. In addition, the exchange between the terminal oxygen atom of $CH_2OO$ and one of the two oxygen atoms in $SO_2$ is necessary for the minor channel forming P2, P3, and P4, consistent with previous research, both experimentally[36] and theoretically[31,32].

The TS energy of the new channel shown at the top of Fig. 3 is 10.38 kcal/mol (CCSD(T)-F12b/cc-pVDZ-F12) or 10.24 kcal/mol (PES, using Newton's method). The wavenumbers ($cm^{-1}$) for the normal modes

of TS are: 58, 111, 196, 292, 369, 410, 469, 551, 729, 973, 1137, 1230, 1293, 1336, 1442, 2378, 2708, and the imaginary frequency is $650i$, obtained at the CCSD-F12b/cc-pVDZ-F12 level of theory.

### Integral cross sections of four products

The integral cross sections ($\sigma$) are calculated by a $b$-weighted numerical integration of the $P(b)$ opacity functions at each collision energy, for each product:

$$\sigma = \pi \cdot \sum_{n=1}^{n_{max}} [b_n - b_{n-1}] \cdot [b_n \cdot P(b_n) + b_{n-1} \cdot P(b_{n-1})]$$

where $b_n = 0.5n$ bohr and $n_{max} = b_{max}/(0.5 \text{ bohr})$ in the present study (see Fig. 2).

Integral cross sections of four products as a function of the collision energy are drawn in Fig. 4. It can be seen that four lines exhibit similar shape even though their values vary greatly. The similar characteristics of the four lines indicates that high-energy collisions are not suitable for the formation of five-member-ring, no matter which product is formed. The integral cross section drops about 6 times from P1 to P2, then drops more than 20 times to P3, and finally P4 rarely appears in our study, because $\cdot OCH_2O \cdot$ is not stable and easily converts to HCOOH or $CO_2 + H_2$.

For the minor channels (P2, P3, and P4), the barrier of the unimolecular reaction from Criegee intermediate to dioxirane is about 20 kcal/mol[51]. However, with the help of $SO_2$ as catalysis, there is no barrier anymore as no threshold energy is found (see Fig. 4).

### Comparison of the old and new channels

During our QCT simulations, a brand new channel that does not experience the five-member-ring complex and forms P1 is observed. The conversion process can be clearly noticed in Fig. 3. This channel clearly shows a direct stripping mechanism. Further exploring the effects of the old and new channels on the integral cross sections for P1 is shown in Fig. 5. The direct stripping channel appears at collision energies higher than 10 kcal/mol, consistent of the TS energy of 10.38 kcal/mol (CCSD(T)-F12b/cc-pVDZ-F12) and 10.24 kcal/mol (PES).

The reaction probabilities for the indirect five-member-ring and direct stripping mechanisms as a function of the impact parameter $b$ are shown in Fig. 6. For the five-member-ring mechanism, two curves almost overlap at the collision energies 20 and 30 kcal/mol, which may indicate that the higher energy does not help to assemble into a ring. For the direct stripping channel however, the reaction probabilities keep growing from 10 to 30 kcal/mol, which fully demonstrates a typical reaction with a barrier. It is reasonable to assume that at higher collision energies the direct mechanism may become important, which needs more experimental confirmation. Considering that the new pathway does not involve the two hydrogen atoms, even if they are substituted by larger groups like methyl or ethyl, the new pathway still exists. In conclusion, we think that the new pathway should be universal for any Criegee intermediates. Of course, additional future studies are needed to reveal its significance.

### Reaction probabilities of the P1 and P2 channels

Reaction probabilities for the P1 and P2 channels corresponding to six different collision energies are plotted as a function of the impact parameter $b$ in Fig. 7. At the low collision energies, i.e., 1 and 2 kcal/mol, reaction probabilities emerge relatively flat in the range of 0 to 8 bohr compared with the steep high collision energy curves. Afterward there is a declining trend; moreover, $b_{max}$ is very large for 1 and 2 kcal/mol (18 and 13 bohr, respectively). At high collision energies from 5 to 30 kcal/mol, the opacity functions almost exhibit rapid decay uniformly and $b_{max}$ is at around 5 to 6 bohr. It is interesting to find the similar patterns for P1 and P2.

The high and broad reaction probability curves at low collision energies show that the $CH_2OO + SO_2$ reaction is very fast at low temperature, which agrees with the common idea that $CH_2OO$ can react with $SO_2$ rapidly.

**Fig. 4 | Comparison of the reactivity of the four products channels.** Integral cross sections as a function of the collision energy for the P1, P2, P3, and P4 channels.

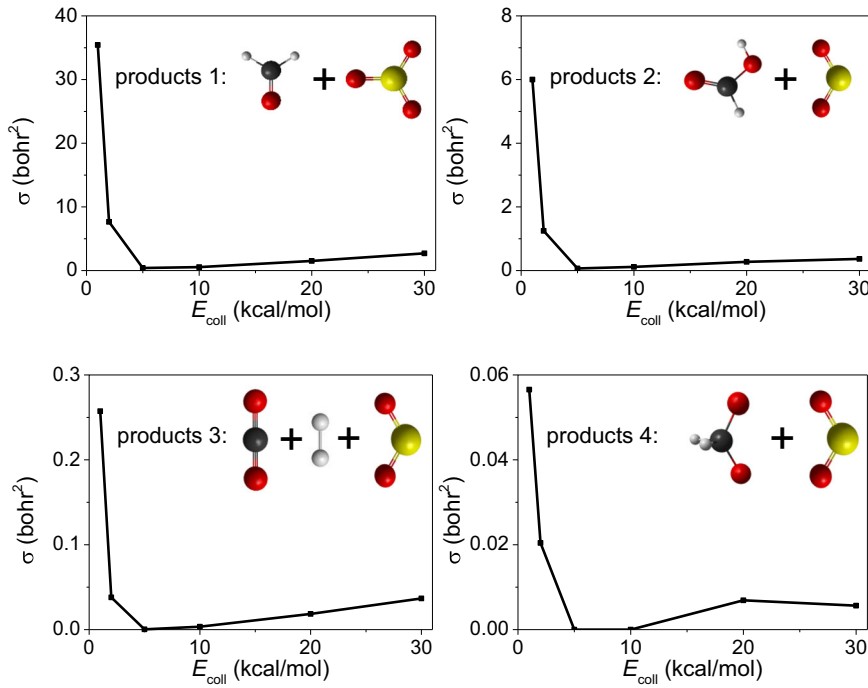

**Fig. 5 | Comparison of the reactivity of direct and indirect mechanism.** Integral cross sections as a function of the collision energy for the indirect and direct mechanism for the P1 channel.

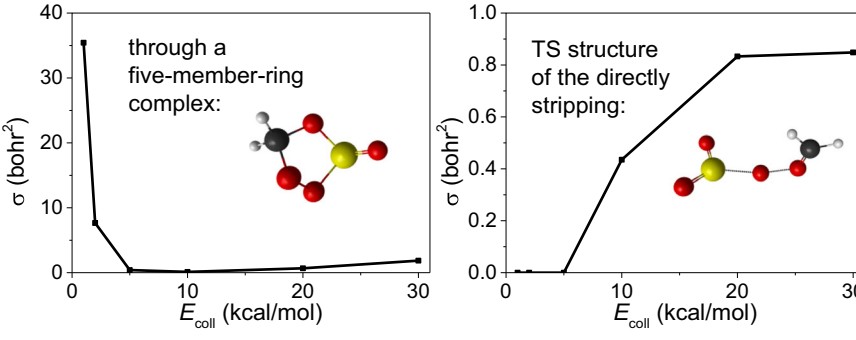

**Fig. 6 | Comparison of the reaction probability of direct and indirect mechanism.** Reaction probabilities calculated with the QCT method for the indirect (conventional pathway, through a five-member-ring complex, left panel) and direct (newly-found pathway, direct stripping process, right panel) mechanisms for the P1 channel at varying collision energies (kcal/mol).

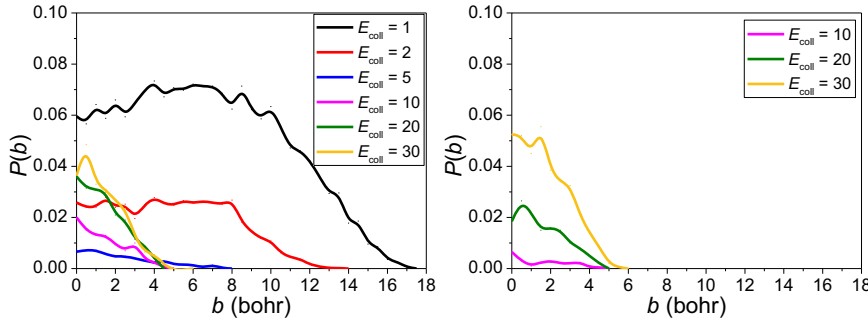

The surprising relatively high reaction probabilities at high collision energies indicate that new complex mechanisms may exist for this reaction. However, we have not found new mechanisms at current stage, which needs additional work in the future.

## Conclusions and outlook

We successfully developed a full-dimensional PES of the $CH_2OO + SO_2$ reaction in this work. First, we selected quantum chemistry computational level appropriate for this system and made necessary tests to guarantee accurate results within a reasonable time. Second, we generated the initial geometry set and our final analytical PES was developed

from 46411 geometries and the corresponding energies in a fifth-order polynomial expansion with the help of ROBOSURFER program package. We compared the PES energies with ab initio values in the entrance and exit regions by 1D-scan and they show a good chemical accuracy for this full-dimensional PES. Third, we performed QCT simulation on this PES and explored dynamical properties of the $CH_2OO + SO_2$ reaction.

Our calculations reported in this paper present four products regarding the $CH_2OO + SO_2$ reaction. P1 ($CH_2O + SO_3$) is the main product channel. To our surprise, a new channel that does not require five-member-ring formation is predicted and only appears when the collision energy is higher than 10 kcal/mol. The threshold value agrees with the TS energy, 10.38 kcal/

**Fig. 7 | Comparison of the reaction probability of P1 and P2 channels.** Reaction probabilities were calculated with the QCT method for P1 (left panel) and P2 (right panel) channels at varying collision energies (kcal/mol).

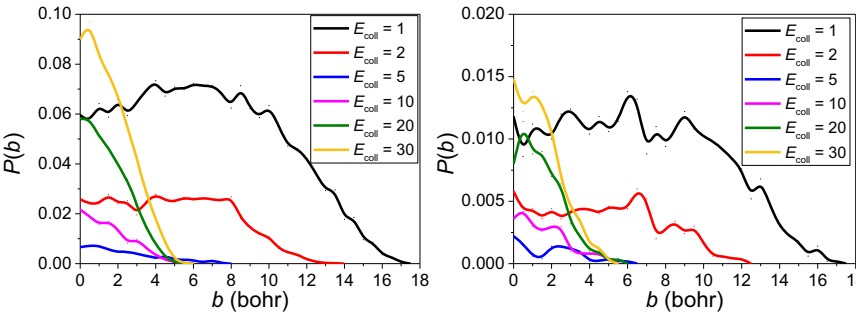

mol (CCSD(T)-F12b/cc-pVDZ-F12) and 10.24 kcal/mol (PES). However, old channels present different behaviors: At low collision energies, reaction probabilities are high and they show flat behavior in the range of 0 to 8 bohr, later show a declining trend and $b_{max}$ reaches large values, i.e., 18 and 13 bohr at $E_{coll} = 1$ and 2 kcal/mol, respectively. It means this reaction can occur very easily, which agrees with the common idea that $CH_2OO$ can react with $SO_2$ very fast. At high collision energies, no constant area appears and the reaction probabilities exhibit rapid decay that reveal a hindered reaction. It inspires us that high temperature may inhibit the $CH_2OO + SO_2$ reaction, which requires us to calculate the reaction rate in the future and will be our next work.

P2 ($HCOOH + SO_2$), P3 ($OCH_2O + SO_2$) and P4 ($CO_2 + H_2 + SO_2$) are minor product channels and the different products appear after three common TSs. Apart from P2 and P4, a new product channel (P3) that was not mentioned in previous work arises in our simulations. This P3 channel promotes us to understand the mechanism in depth. The integral cross sections for P1, P2, P3, and P4 decrease as the collision energies increase indicating that high-energy collisions are not suitable for the formation of the five-member-ring. Our global analytical PES and the QCT simulation can provide insight into reaction channels, enable further characterization of this important class of Criegee intermediates related reactions, and hopefully motivate further experimental and theoretical studies on this reaction family.

## Data availability

Optimized Cartesian coordinates (in angström) and energies (in atomic units) obtained on the PES are available in Supplementary Data 1.

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

## Acknowledgements

We thank Tibor Győri and Viktor Tajti for their tips and discussions. This work was supported by the National Research, Development and Innovation Office−NKFIH, K-125317 and K-146759; Project no. TKP2021-NVA-19, provided by the Ministry of Innovation and Technology of Hungary from the National Research, Development and Innovation Fund, financed under the TKP2021-NVA funding scheme; and the Momentum (Lendület) Program of the Hungarian Academy of Sciences. Article processing charge was covered by the University of Szeged Open Access Fund (grant number 7136).

## Author contributions

Cangtao Yin: conceptualization, investigation, analysis, writing. Gábor Czakó: writing-review, supervision. Correspondence to Cangtao Yin or Gábor Czakó.

## Funding

## Competing interests

The authors declare no conflicts of interest.
