## [Peer Review File · Communications Chemistry]

Referee reports: first round

Reviewers' comments:

Reviewer #1 (Remarks to the Author):

This study presents some interesting new pathway and mechanism for the reaction of CH₂OO with SO₂. The title reaction has important implications for atmospheric oxidation of SO₂ and it's great to see a theoretical study with dynamical insight. The stripping mechanism for production of SO₃ has not been suggested previously and maybe important at higher temperatures. I recommend publishing this study once the authors have answered the major comment and made the other minor changes.

Major comment, previous authors such as Vereecken and Kuwata have pointed out that some of the stationary points on the 'minor' reaction pathway have biradical character and multi-reference methods were used to calculate more accurate energies. Some discussion about is needed here to justify the use of only single reference methods in this study.

Minor comments, there are quite a few instances where I think the quality of writing needs to be cross checked by the editor. Here are some that were obvious to me.

- 1) In page 3, Table 1, citation to ref 22 needs to be corrected.
- 2) Page 3, line 94, change 'chemical' to 'chemically'
- 3) Page 3, line 106, change 'made use of' to 'used to'
- 4) Page 4, line 118, citation to ref 44 needs to be corrected.
- 5) Page 4, line 121, change 'keeping to pay attention to' to 'using'
- 6) Page 4, line 143, delete 'is'
- 7) Page 5, Figure 1, make the data points in the plots bigger
- 8) Page 5, line 181, change 'offers' to 'shows'
- 9) Page 7, line 238, delete 'extremely'
- 10) Page 10, line 292, change 'develop' to 'developed'
- 11) Page 10, line 299, change 'perfect' to 'good'
- 12) Page 10, line 299, change 'perform' to 'We performed'

Reviewer #2 (Remarks to the Author):

On the basis of potential energy surface construction and quasi-classical trajectory (QCT) simulations, the authors studied the dynamics of the CH₂OO+SO₂ reaction. A new pathway (direct stripping) forming the main product (CH₂OO+SO₃) is discovered. This manuscript can be recommended to be published in COMMSCHEM after the following issues being properly addressed.

1. When discussing the new pathway, analyses about free energy barrier and rate constant are desirable.
2. How about the effect of the concentration of reactants?
3. CH₂OO is the simplest Criegee intermediate, is the new pathway found in this study a universal one for larger Criegee intermediates? If not, this work does not make a significant contribution to the field.
4. Figure 1 should be replaced by a table

Referee reports: first round

Reviewer #3 (Remarks to the Author):

Communications Chemistry - Reviewer Report

Manuscript COMMSCHEM-24-0034-T

Title: Revealing new pathways for the reaction of Criegee intermediate CH_2OO with SO_2

This manuscript reports on a theoretical study of the reaction between CH_2OO and SO_2 . The authors produce a full-dimensional potential energy surface for the reaction and run quasi-classical trajectories. They tested different ab initio methods to find a suitable method that balances accuracy and efficiency. The major result was the discovery of a claimed new reaction pathway that leads to the major products $\text{SO} + \text{CH}_2\text{O}$, as well as a new minor pathway that leads to $\text{CO}_2 + \text{H}_2 + \text{SO}_2$. This looks to be a complicated study, and the results are potentially interesting, but the manuscript lacks a great deal of detail and explanation that is integral to understanding. Due to the lack of details and the surface level analysis of the results, I would recommend rejection and resubmission following a major revision that addresses the below comments.

Comments:

1. I recommend that the authors do a more thorough literature search, as they fail to cite a few important Criegee intermediate reaction papers, for example the paper by the

Taatjes/Lester/Klippenstein groups in PNAS regarding MVK-Oxide + SO_2

(<http://www.pnas.org/lookup/doi/10.1073/pnas.1916711117>) .

2. PIMS is actually MPIMS (Multiplexed Photoionization Mass Spectrometry).

3. The "heteroozonide" that is the 5 membered ring is usually referred to as a secondary ozonide (SOZ).

4. The authors should comment in the introduction that the main atmospheric removal mechanism for CH_2OO is reaction with the water dimer, so the $\text{CH}_2\text{OO} + \text{SO}_2$ reaction will not have an impact on Criegee atmospheric chemistry.

5. I am not aware of the catalytic 1,2-H atom shift mechanism with SO_2 . Can the authors provide more info to me about this? Also, please change the term to "1,2-H atom shift".

6. Line 51, I would not use the term "discovered", I recommend "predicted".

7. Line 97: It is not surprising that adding the perturbative triple excitations to the CCSD method produces a more accurate energy, and that an increase from the DZ to TZ basis set would produce a smaller increase in accuracy. For the energies given in Table 1, are the geometries optimized at the stated level of theory, or is it a single-point calculation at a single geometry?

8. What is the CCSD(T) T1 diagnostic for each of these calculations?

9. I think the choice of the DZ basis set is sufficient given the increased accuracy of the F12b methodology.

10. Reading Section II.B., it seems like the PES points are at the CCSD(T)-F12b/cc-pVDZ-F12 // MP2/aug-cc-pVDZ. Is this correct? For the points that failed, what was the source of the error? Similar to comment 8, are the T1 diagnostic values reasonable for all of the points? Have they been checked for all 17, 245 computations?

11. For the "holes" in the PES, how does the ROBOSURFER program fill in these points?

12. Line 147 the authors use aug-cc-pVDZ but previous aVDZ. Please introduce the full basis set the first time, then use the abbreviation.

Referee reports: first round

13. Line 116, what does the a value represent in the function, and how was 2 bohr chosen?
14. The reference for the NH_3 study is incorrect, it should be 46.
15. How is the PES function of $y = \exp(-r/a)$ chosen? And is there benchmarking done to show that it is a good function to use for fitting this type of PES? I am not familiar with what goes into choosing these functions.
16. Line 121, number 2) I am not sure what this sentence is trying to say. What does it keep track of? And what new geometries is it producing?
17. What is the "comprehensive inspection" that is done on the data before they are added to the dataset?
18. How long does each single-point calculation take at the CCSD(T)-F12b level?
19. I am a little bit confused by Section II C and D. In Section C it is stated that there are 19,500 geometries, 187 fail (but the reason is not stated), and then after removal of high-energy geometries, they are left with 17,245 points. Then in Section D, they have 38,716 geometries after 418 iterations.
 - a. What is the discrepancy between the number of geometries?
 - b. Why do some geometries fail?
 - c. How is the PES iteratively improved?
20. Why did the authors use a cutoff of 400 kcal/mol? And why do they believe this belongs to the $\text{HCOOH} + \text{SO}_2$ products? In Table 1 this product is at -116 kcal/mol.
21. With 10,220 fitting coefficients, how is the fit evaluated and ensured it is good? I would think you could fit anything with 10,220 coefficients, so why is it well-done?
22. In Section II D, the final number of points in the PES is now up to 46,411 after failed CCSD calculations lowered it to 37,676. Can the authors elaborate on this further?
23. Line 154, what do the authors mean by "188 iterations are conducted to lower the percentage of unphysical trajectories to less than 1%"? Are the authors producing the PES here or running trajectories? And again, what are the iterations doing?
24. In Fig. 1, are these plots taken from the PES or newly calculated? It says from the PES but in the text above it describes how they are computed.
25. Why are the 1-D scans performed if the PES is full-dimensional?
26. What is meant by the asymptote of the complex exo? Is this a transition state, minimum, or asymptote?
27. How are the trajectories run? For example: What is the step size? which algorithms are used for the classical part? Where does the ZPE in the trajectories come from? Are there any quantum effects included in the trajectories? How long are the trajectories run for? These are just a few questions. I ask the authors to please include details about the trajectories.
28. Do you get branching ratios from the trajectories? The authors mention "most cases, some cases, a few cases" for the pathways. Can you give a quantitative measure?
29. How is the collisional energy implemented in the trajectories? Is this effectively a temperature? With some of the pathways having a large TS barrier, I would expect even the trajectories with 30 kcal/mol energies to have trouble completing.
30. What is an integral cross section and how is it determined?
31. It is very surprising to me that the integral cross sections (which I presume equate to reaction probability or reaction rate) increase with energy. It is well-known that the rate of bimolecular reaction has a negative temperature dependence due to the need to interact for a longer time in order to initiate chemical changes. Why in all of these cases is a higher energy (and then higher temperature) leading to more reactivity? And why is the cross section so much higher at the lowest collision energy? I would expect a very small energy to not lead to reaction.

Referee reports: first round

32. How is the TS energy determined by the PES? And what are the frequencies for this TS that you got from the ab initio calculations?

33. In the Fig 4 caption, what does "direct" and "indirect" mechanism refer to?

34. Line 270, "It is reasonable to assume that at higher collision energies the direct mechanism may become important," does this have any implications to experiment or the atmosphere? Is there a direct connection between collision energy and a physical process like temperature?

35. Is Fig 4 Left panel and Fig. 5 Left panel the same data? Why did the authors choose one pathway vs the other?

36. The analysis for the probability plots is lacking. The authors describe the shapes of the curves, but do not go into any analysis or chemical implications of this.

37. The probability curves look very similar to me at all energies. Does this mean that the reactions have similar TS barriers? Or that they will compete?

38. The authors present a vast amount of quantum chemical data in this paper but do not include any of it in an appendix or supplementary information. To properly scrutinize this work, the geometries, energies, vibrational frequencies, T1 diagnostics, and other quantum chemical data should be provided to the reviewer and the audience. The statement that it is available by request is not sufficient.

39. The authors mention the new pathway to form $\text{CO}_2 + \text{H}_2 + \text{SO}_2$, but do not give a mechanism, energies, barriers, implications, etc. They say that an advantage of QCT is you get the geometries and can follow the reaction. If this is new and interesting, why did they not show a mechanism or delve deeper into the reaction? In the conclusion they write "This novel P3 channel promotes us to understand the mechanism in depth." But they do not discuss the mechanism or if it is even possible.

40. Some interesting points are discussed in the conclusions that would fit very nicely if placed in the discussion section and expanded upon. An example of which is "However, old channels present different behaviors: At low collision energies, reaction probabilities are high and they show flat behavior in the range of 0 to 8 bohr, later show a declining trend and bmax reaches largest values, i.e., 18 and 13 bohr at $E_{\text{coll}} = 1$ and 2 kcal/mol, respectively. It means this reaction can occur very easily, which agrees with the common idea that CH_2OO can react with SO_2 very fast."

Response to Referee 1

This study presents some interesting new pathway and mechanism for the reaction of CH₂OO with SO₂. The title reaction has important implications for atmospheric oxidation of SO₂ and it's great to see a theoretical study with dynamical insight. The stripping mechanism for production of SO₃ has not been suggested previously and maybe important at higher temperatures. I recommend publishing this study once the authors have answered the major comment and made the other minor changes.

Reply: We thank the Reviewer for the careful review and positive recommendation.

Major comment, previous authors such as Vereecken and Kuwata have pointed out that some of the stationary points on the 'minor' reaction pathway have biradical character and multi-reference methods were used to calculate more accurate energies. Some discussion about is needed here to justify the use of only single reference methods in this study.

Reply: We have added the following discussion about the validity of single reference methods on page 4, highlighted in green color.

"Single reference methods are safe for all the exit channels since the T_1 values of the products are 0.015 (CH₂O), 0.016 (HCOOH) and 0.027 (OCH₂O), respectively. The one with more multi-reference character is the reactant CH₂OO, whose T_1 value is 0.044 (Vereecken et al.³¹) or 0.042 (our calculation), which is right on the dividing line between whether one must use multireference methods or not. Although some of the stationary points have higher T_1 values (e.g., TS-12a in Kuwata et al.³²), the most important geometries, i.e., the five-member-ring heteroozonide complex *endo* and *exo*, have T_1 values below 0.02, which are low enough to be treated with a single-reference method. In addition, as Yin and Takahashi⁴⁴ investigated, ignoring the static correlation only causes an underestimation of the TS energy by about 1 kcal/mol in the unimolecular reaction of CH₂OO. Vereecken et al.⁴⁵ and Anglada et al.⁴⁶ also confirmed that some other Criegee related systems can be treated with single-reference methods. As a conclusion, although Criegee reactions have some multireference character at certain regions of the configuration space, in practice the use of single-reference methods is sufficient for this system."

Minor comments, there are quite a few instances where I think the quality of writing needs to be cross checked by the editor. Here are some that were obvious to me.

- 1) In page 3, Table 1, citation to ref 22 needs to be corrected.
- 2) Page 3, line 94, change 'chemical' to 'chemically'
- 3) Page 3, line 106, change 'made use of' to 'used to'
- 4) Page 4, line 118, citation to ref 44 needs to be corrected.
- 5) Page 4, line 121, change 'keeping to pay attention to' to 'using'
- 6) Page 4, line 143, delete 'is'
- 7) Page 5, Figure 1, make the data points in the plots bigger
- 8) Page 5, line 181, change 'offers' to 'shows'
- 9) Page 7, line 238, delete 'extremely'

10) Page 10, line 292, change 'develop' to 'developed'

11) Page 10, line 299, change 'perfect' to 'good'

12) Page 10, line 299, change 'perform' to 'We performed'

Reply: We have corrected these mistakes and double-checked the whole manuscript to improve the quality of writing.

Response to Referee 2

On the basis of potential energy surface construction and quasi-classical trajectory (QCT) simulations, the authors studied the dynamics of the $\text{CH}_2\text{OO}+\text{SO}_2$ reaction. A new pathway (direct stripping) forming the main product ($\text{CH}_2\text{OO}+\text{SO}_3$) is discovered. This manuscript can be recommended to be published in COMMSCHEM after the following issues being properly addressed.

Reply: We thank the Reviewer for the careful review and positive recommendation.

1. When discussing the new pathway, analyses about free energy barrier and rate constant are desirable.

Reply: In this work our main focus is about PES development and the QCT simulations on the PES. The vibrational and rotational energies are set to their zero-point values as our initial condition, and the translational energy (collision energy) is from 1 to 30 kcal/mol. This special condition does not correspond to any real temperature. Thus no free energy can be deduced. As for the rate constant, it will be our next step of studying this system.

2. How about the effect of the concentration of reactants?

Reply: The QCT simulation does not involve concentration, as it studies chemical reactions under single collision conditions, like crossed-beam experiments. Thus, it provides reaction probabilities, cross sections, etc. by simulating tens of thousands of collisions between two molecules.

3. CH_2OO is the simplest Criegee intermediate, is the new pathway found in this study a universal one for larger Criegee intermediates? If not, this work does not make a significant contribution to the field.

Reply: We have added the following discussion on the possible universal feature of the new pathway on page 11, highlighted in blue color.

“Considering that the new pathway does not involve the two hydrogen atoms, even if they are substituted by larger groups like methyl or ethyl, the new pathway still exists. In conclusion, we think that the new pathway should be universal for any Criegee intermediates. Of course, additional future studies are needed to reveal its significance.”

4. Figure 1 should be replaced by a table.

Reply: We think that the Reviewer meant Scheme 1 when they wrote Figure 1, because Figure 1 shows potential energy curves, which cannot be replaced by a table. Scheme 1 shows the QCT initial conditions, which could also be given in a table as the Reviewer suggested. However, in order to capture the attention of the broad readership of Communications Chemistry, we think it is better to represent the initial conditions in a scheme rather than in a table.

Response to Referee 3

This manuscript reports on a theoretical study of the reaction between CH₂OO and SO₂. The authors produce a full-dimensional potential energy surface for the reaction and run quasi-classical trajectories. They tested different ab initio methods to find a suitable method that balances accuracy and efficiency. The major result was the discovery of a claimed new reaction pathway that leads to the major products SO₃ + CH₂O, as well as a new minor pathway that leads to CO₂ + H₂ + SO₂. This looks to be a complicated study, and the results are potentially interesting, but the manuscript lacks a great deal of detail and explanation that is integral to understanding. Due to the lack of details and the surface level analysis of the results, I would recommend rejection and resubmission following a major revision that addresses the below comments.

Reply: We thank the Reviewer for the careful review and we have addressed their comments as detailed below.

Comments:

1. I recommend that the authors do a more thorough literature search, as they fail to cite a few important Criegee intermediate reaction papers, for example the paper by the Taatjes/Lester/Klippenstein groups in PNAS regarding MVK-Oxide + SO₂ (<http://www.pnas.org/lookup/doi/10.1073/pnas.1916711117>) .

Reply: There are so many important Criegee intermediate reaction papers, we have cited the ones that involve CH₂OO + SO₂ since our work is about this simplest Criegee intermediate. And, of course, we already cited many papers from Taatjes/Lester/Klippenstein groups, for example, Refs. 2, 4, 5, 9, 13, 19. Nevertheless, we have added the above-mentioned PNAS paper to the revised manuscript as Ref. 18.

2. PIMS is actually MPIMS (Multiplexed Photoionization Mass Spectrometry).

Reply: When we mentioned PIMS, we cited the work of Howes et al.,²⁶ where the authors wrote “photoionization mass spectrometry” instead of “Multiplexed Photoionization Mass Spectrometry”.

3. The “heteroozonide” that is the 5 membered ring is usually referred to as a secondary ozonide (SOZ).

Reply: We agree with the Reviewer that the 5 membered ring is also referred to as SOZ. However, since we are using the geometries of Kuwata et al.,³² and they called it heteroozonide, we followed their name for consistency.

4. The authors should comment in the introduction that the main atmospheric removal mechanism for CH₂OO is reaction with the water dimer, so the CH₂OO + SO₂ reaction will not have an impact on Criegee atmospheric chemistry.

Reply: According to the Referee’s suggestion, we have added the following sentence on page 2, highlighted in red color:

“Although the main atmospheric removal mechanism for CH₂OO is the reaction with the water dimer, the negative temperature dependence for the CH₂OO + SO₂

reaction in conjunction with the decrease in water dimer concentration at lower temperatures means that CH₂OO can play an enhanced role in SO₂ oxidation in the atmosphere at lower temperatures.²⁸ In addition, some Criegee intermediates can survive in humid condition, for example, methacrolein oxide, an isoprene-derived Criegee intermediate.¹⁶”

5. I am not aware of the catalytic 1,2-H atom shift mechanism with SO₂. Can the authors provide more info to me about this? Also, please change the term to “1,2-H atom shift”.

Reply: The 1,2-H atom shift mechanism is more clear with the help of reaction (3) in Ref. 31, i.e., one of the H atoms is shifted from C atom to O atom. And we have deleted the term “1,2-H shift” in our manuscript.

6. Line 51, I would not use the term “discovered”, I recommend “predicted”.

Reply: We have changed the term as the Reviewer recommended.

7. Line 97: It is not surprising that adding the perturbative triple excitations to the CCSD method produces a more accurate energy, and that an increase from the DZ to TZ basis set would produce a smaller increase in accuracy. For the energies given in Table 1, are the geometries optimized at the stated level of theory, or is it a single-point calculation at a single geometry?

Reply: We have made it more clear in Table 1, highlighted in red color:

“For the MP2/aVDZ and CCSD-F12b/cc-pVDZ-F12 energies, the geometries are optimized at the stated level of theory; for the CCSD(T)-F12b/cc-pVDZ-F12 and CCSD(T)-F12b/cc-pVTZ-F12 energies, the geometries are optimized at the CCSD-F12b/cc-pVDZ-F12 level.”

8. What is the CCSD(T) T₁ diagnostic for each of these calculations?

Reply: We have added the T₁ diagnostics on page 4, highlighted in red color:

“At the CCSD-F12b/cc-pVDZ-F12 level of theory, the T₁ diagnostic values are 0.0178 and 0.0183 for complex *endo* and complex *exo*, respectively, whereas the corresponding T₁ values are 0.0179 and 0.0184 with the larger cc-pVTZ-F12 basis set, in order.”

9. I think the choice of the DZ basis set is sufficient given the increased accuracy of the F12b methodology.

Reply: We agree that DZ basis set is sufficient so we chose CCSD(T)-F12b/cc-pVDZ-F12 as our calculation level in this manuscript to guarantee the accuracy and efficiency. To make the reason of the fast basis-set convergence clear, we write on page 3, highlighted in red color:

“As for the coupled-cluster theory, a large difference occurs between CCSD-F12b/cc-pVDZ-F12 and CCSD(T)-F12b/cc-pVDZ-F12, while a small one happens between CCSD(T)-F12b/cc-pVDZ-F12 and CCSD(T)-F12b/cc-pVTZ-F12 due to the fast basis-set convergence of the explicitly-correlated F12b method.”

10. Reading Section II.B., it seems like the PES points are at the CCSD(T)-F12b/cc-pVDZ-F12 // MP2/aug-cc-pVDZ. Is this correct? For the points that failed, what was the source of the error? Similar to comment 8, are the T1 diagnostic values reasonable for all of the points? Have they been checked for all 17, 245 computations?

Reply: At first, the energies are calculated at the MP2/aug-cc-pVDZ level, since it is fast to get more geometry points. Then we recalculated them at CCSD(T)-F12b/cc-pVDZ-F12. The 31 stationary points are taken from Ref. 32, where the geometries were optimized at B3LYP/6-311(2d,d,p). All the other geometries are generated by initial random displacements and chosen from trajectories.

We have revised the reasons for the failed geometries on page 5, highlighted in red color:

“Only 187 geometries failed due to HF convergence issues which can be considered negligible.”

“2.7% of the points (around one thousand) failed in the coupled-cluster iterations and 37,676 points are well reserved.”

As we discussed below Table 1, although some geometries have higher T_1 values, the exit channels and the five-member-ring heteroozonide complex have low T_1 values and can be treated with single-reference method. The one with more multi-reference character is the reactant CH₂OO, whose T_1 value is 0.044 (Vereecken et al.³¹) or 0.042 (our calculation). However, as Yin and Takahashi⁴⁴ investigated, ignoring the static correlation only causes an underestimation of the TS energy by about 1 kcal/mol in the unimolecular reaction of CH₂OO.

11. For the “holes” in the PES, how does the ROBOSURFER program fill in these points?

Reply: We have added the explanation on page 4, highlighted in red color:

“It picks the geometries along the trajectory before it fails, i.e., goes into a hole. Then the program calculates the energy, evaluates its usefulness, and then adds it into the current geometry set. After hundreds of iterations, very few trajectories fail and most holes are filled in.”

12. Line 147 the authors use aug-cc-pVDZ but previous aVDZ. Please introduce the full basis set the first time, then use the abbreviation.

Reply: We have fixed this issue.

13. Line 116, what does the α value represent in the function, and how was 2 bohr chosen?

Reply: We have added the explanation on page 4, highlighted in red color:

“ α is a parameter that controls the asymptotic behaviour of the PES. Our group have tested it and found out that 2 bohr is usually a good choice for neutral systems, including a similar system CH₂OO + NH₃.⁴⁸”

14. The reference for the NH₃ study is incorrect, it should be 46.

Reply: We have fixed this mistake, and now it is Ref. 48 in the revised manuscript.

15. How is the PES function of $y = \exp(-r/a)$ chosen? And is there benchmarking done to show that it is a good function to use for fitting this type of PES? I am not familiar with what goes into choosing these functions.

Reply: We have added the validation of this function form on page 5, highlighted in red color:

“The $\exp(-r_{ij}/a)$ function is a common choice when fitting the potential energy vs distance of two atoms, since its second-order polynomial expansion behaves like Morse potential and it guarantees that the interaction energy goes to zero when two atoms are far away from each other.”

16. Line 121, number 2) I am not sure what this sentence is trying to say. What does it keep track of? And what new geometries is it producing?

Reply: We have changed the phrase “keep track of” to “using”. The new geometries come from the trajectories of the simulation.

17. What is the “comprehensive inspection” that is done on the data before they are added to the dataset?

Reply: We have extended the discussion on page 5, highlighted in red color:

“The subprogram of ROBOSURFER, called ADDPOINTS, adds geometries into the fitting set. It tries to reduce the scaled fitting error for all geometries while keeping the fitting set as small as possible. More details can be found in ref. 39.”

18. How long does each single-point calculation take at the CCSD(T)-F12b level?

Reply: We have added the required time on page 4, highlighted in red color:

“which takes around ten minutes for each *ab initio* calculation at the level of CCSD(T)-F12b/cc-pVDZ-F12.”

19. I am a little bit confused by Section II C and D. In Section C it is stated that there are 19,500 geometries, 187 fail (but the reason is not stated), and then after removal of high-energy geometries, they are left with 17,245 points. Then in Section D, they have 38,716 geometries after 418 iterations.

a. What is the discrepancy between the number of geometries?

Reply: As we explained on page 5, in section C, we built the initial PES from 17,245 points, and then in section D the PES is further improved by adding more points. The final coupled-cluster PES is therefore developed from 46,411 geometries.

b. Why do some geometries fail?

Reply: We have revised the reasons for the failed geometries on page 5, highlighted in red color:

“Only 187 geometries failed due to HF convergence issues which can be considered negligible.”

“2.7% of the points (around one thousand) failed in the coupled-cluster iterations and 37,676 points are well reserved.”

c. How is the PES iteratively improved?

Reply: As we explained in Section D, page 4 and 5, the main function of ROBOSURFER is adding new geometries into the geometry set thus improving the PES.

20. Why did the authors use a cutoff of 400 kcal/mol? And why do they believe this belongs to the HCOOH + SO₂ products? In Table 1 this product is at -116 kcal/mol.

Reply: We have added the explanation on page 5, highlighted in red color:

“Since some geometries are very distorted with super high energies, they can be harmful to the analytical PES. It is necessary to exclude those geometries so a critical value of energy above the global minimum (product) should be set. In our experience 400 kcal/mol is usually a good choice.”

It does not have to belong to the product. Any geometry with twisted part can have a high energy. The limitation of 400 kcal/mol is relative to the global minimum, and the global minimum corresponds to the energy of minor products HCOOH + SO₂.

21. With 10,220 fitting coefficients, how is the fit evaluated and ensured it is good? I would think you could fit anything with 10,220 coefficients, so why is it well-done?

Reply: The CH₂OO + SO₂ system contains 8 atoms so 18 dimensions. It is quite a large system and actually not easy to get a good analytical PES. Furthermore, we note that we have about 4.5-times more points than coefficients.

The evaluation of PES is displayed on page 6, including 1-D scan (Fig. 1) and the stationary-point energies (Table 2), which show good agreement compared to the *ab initio* reference results. More importantly, section IV (page 8 to 11) shows that the QCT simulations on the PES reproduced the right reaction pathways from previous research, both experimental³⁶ and theoretical,^{31,32} for this reaction.

22. In Section II D, the final number of points in the PES is now up to 46,411 after failed CCSD calculations lowered it to 37,676. Can the authors elaborate on this further?

Reply: As we explained in Section II D, actually the CCSD calculations lowered the number of points from 38,716 to 37,676, only 2.7% failed. After that, we launched ROBOSURFER again and the number of points reached 46,411, which is the final geometry set.

23. Line 154, what do the authors mean by “188 iterations are conducted to lower the percentage of unphysical trajectories to less than 1%”? Are the authors producing the PES here or running trajectories? And again, what are the iterations doing?

Reply: As we explained in Section II B, the program ROBOSURFER runs trajectories on the current version of the PES to generate new geometries. After adding more points into the geometry set a new PES can be fitted, which should be better than the previous one. That is an iteration in ROBOSURFER.

24. In Fig. 1, are these plots taken from the PES or newly calculated? It says from the PES but in the text above it describes how they are computed.

Reply: The solid lines are from PES and the dotted lines are calculated in Fig. 1. By comparing the difference between them, we can evaluate the PES. This philosophy also applies for Table 2, see the highlighted sentence on page 6:

“obtained from the analytical PES and CCSD(T)-F12b/cc-pVDZ-F12 ab initio calculations.”

The text above Fig. 1 describes how these geometries are generated.

25. Why are the 1-D scans performed if the PES is full-dimensional?

Reply: We have added the following explanation on page 6, highlighted in red color:

“To show the performance of the full-dimensional PES, we adapted the 1-D scan method, i.e., elongating and shortening the C-S bond of an optimized geometry, while keeping the other degrees frozen, to evaluate the accuracy of the PES. The results are shown in Fig. 1.”

26. What is meant by the asymptote of the complex *exo*? Is this a transition state, minimum, or asymptote?

Reply: On page 6 we described how the 1-D scan geometries are generated:

“The C-S bond of the optimized complex *exo* geometry is elongated and shortened, while other degrees of the two parts are frozen.”

Thus the asymptote of the complex *exo* is not a stationary point. It is an artificial geometry just to evaluate the performance of the PES.

27. How are the trajectories run? For example: What is the step size? which algorithms are used for the classical part? Where does the ZPE in the trajectories come from? Are there any quantum effects included in the trajectories? How long are the trajectories run for? These are just a few questions. I ask the authors to please include details about the trajectories.

Reply: We have extended the details about the trajectories on page 7, highlighted in red color:

“The initial condition gives the positions and velocities for each atom, in a way that the energies of the reactants correspond to their ZPE values. The forces are obtained from the PES by computing numerical gradients. Then the positions and velocities of the next step (0.0726 fs) can be calculated using the velocity-Verlet algorithm. Besides ZPE, no other quantum effects are included in the process of simulations. If the distance between the farthest atoms is greater than the largest initial one by 1 bohr, the propagation will be terminated.”

28. Do you get branching ratios from the trajectories? The authors mention “most cases, some cases, a few cases” for the pathways. Can you give a quantitative measure?

Reply: Yes. We got the branching ratios from trajectories. The quantitative description is displayed in Fig. 2 to 5.

29. How is the collisional energy implemented in the trajectories? Is this effectively a temperature? With some of the pathways having a large TS barrier, I would expect even the trajectories with 30 kcal/mol energies to have trouble completing.

Reply: We have explained the collisional energy implementation on page 7, highlighted in red color:

“The collision energy is the initial X-direction translational energy in the trajectories, i.e., the relative motion energy of the two reactants, CH₂OO and SO₂. It is not a temperature because the energy corresponding to a temperature should be distributed among the three directions of translation, three rotations, and $3N - 6$ vibrations, where N is the number of the atoms.”

It actually does not matter how large the TS barrier is for the completion of trajectories. If it is larger than the collision energy, no reaction can occur; the two reactants collide and then separate until they are far from each other. If it is smaller than the collision energy, there is a chance for reaction, depending on lots of factors, for example, the orientations of the reactants, the energy distribution in the reactants, etc. After thousands of trajectories, we have the statistically reaction probability shown in Fig. 4 and 5.

In addition, we should clarify that most of the pathways in this reaction are barrierless. The only reaction pathway with a barrier of around 10 kcal/mol is the newly-found direct stripping mechanism, shown in the right panel of Fig. 3, which only happens when the collision energy is above 10 kcal/mol, in consistency to the TS barrier.

30. What is an integral cross section and how is it determined?

Reply: We present the formula for calculating the integral cross section on page 9, highlighted in red color:

“The integral cross sections (σ) are calculated by a b -weighted numerical integration of the $P(b)$ opacity functions at each collision energy, for each product:

$$\sigma = \pi \cdot \sum_{n=1}^{n_{\max}} [b_n - b_{n-1}] \cdot [b_n \cdot P(b_n) + b_{n-1} \cdot P(b_{n-1})] ,$$

where $b_n = 0.5n$ bohr and $n_{\max} = b_{\max}/(0.5 \text{ bohr})$ in the present study (see Scheme 1).”

31. It is very surprising to me that the integral cross sections (which I presume equate to reaction probability or reaction rate) increase with energy. It is well-known that the rate of bimolecular reaction has a negative temperature dependence due to the need to interact for a longer time in order to initiate chemical changes. Why in all of these cases is a higher energy (and then higher temperature) leading to more reactivity? And why is the cross section so much higher at the lowest collision energy? I would expect a very small energy to not lead to reaction.

Reply: Yes, it was interesting for us too that the integral cross sections increase with collision energy, and we have added the following discussion on page 12, highlighted in red color:

“The surprising relatively high reaction probabilities at high collision energies indicate that new complex mechanisms may exist for this reaction. However, we have

not found new mechanisms at current stage, which needs additional work in the future.”

As shown in Fig. 5, b_{\max} is very large (18 and 13 bohr) at low collision energies (1 and 2 kcal/mol), which leads to much higher cross sections at low collision energies. Since it is a barrierless reaction, even very small energy can lead to reaction.

32. How is the TS energy determined by the PES? And what are the frequencies for this TS that you got from the *ab initio* calculations?

Reply: We have added the following frequencies for the new TS on page 9, highlighted in red color:

“The TS energy of the new channel shown on the top of Scheme 2 is 10.38 kcal/mol (CCSD(T)-F12b/cc-pVDZ-F12) or 10.24 kcal/mol (PES, using Newton’s method). The wavenumbers (cm^{-1}) for the normal modes of TS are: 58, 111, 196, 292, 369, 410, 469, 551, 729, 973, 1137, 1230, 1293, 1336, 1442, 2378, 2708, and the imaginary frequency is $650i$, obtained at the CCSD-F12b/cc-pVDZ-F12 level of theory.”

33. In the Fig 4 caption, what does “direct” and “indirect” mechanism refer to?

Reply: We have extended the explanation in the Fig. 4 caption, on page 11, highlighted in red color:

“indirect (conventional pathway, through a five-member-ring complex, left panel) and direct (newly-found pathway, direct stripping process, right panel) mechanisms”

34. Line 270, “It is reasonable to assume that at higher collision energies the direct mechanism may become important,” does this have any implications to experiment or the atmosphere? Is there a direct connection between collision energy and a physical process like temperature?

Reply: We have added the implications to experiment and atmosphere on page 11, highlighted in red color:

“which needs experimental confirmation and atmospheric detection.”

For the connection between collision energy and temperature, as we explained for question 29:

“The collision energy is the initial X-direction translational energy in the trajectories, i.e., the relative motion energy of the two reactants, CH_2OO and SO_2 . It is not a temperature because the energy corresponding to a temperature should be distributed among the three directions of translation, three rotations, and $3N - 6$ vibrations, where N is the number of the atoms.”

Thus, high collision energy has similar impact as high temperature to some extent.

35. Is Fig 4 Left panel and Fig. 5 Left panel the same data? Why did the authors choose one pathway vs the other?

Reply: There are two mechanisms for P1, indirect and direct mechanism, shown in the left and right panel of Fig. 4, for comparison. This is explained in more details in the revised manuscript. Fig. 5 shows P1 in left panel and P2 in right panel. By adding the data of both panels in Fig. 4, one can get the data of the left panel in Fig. 5.

36. The analysis for the probability plots is lacking. The authors describe the shapes of the curves, but do not go into any analysis or chemical implications of this.

Reply: We have extended this description on page 12, highlighted in red color:

“The high and broad reaction probability curves at low collision energies show that the $\text{CH}_2\text{OO} + \text{SO}_2$ reaction is very fast at low temperature, which agrees with the common idea that CH_2OO can react with SO_2 rapidly.”

37. The probability curves look very similar to me at all energies. Does this mean that the reactions have similar TS barriers? Or that they will compete?

Reply: We do not consider the probability curves similar with each other at different energies. Take the left panel of Fig. 4 for example, at low energy the curve is very high and broad ($E_{\text{coll}} = 1$ kcal/mol, black line), while at high energy ($E_{\text{coll}} > 10$ kcal/mol) the curve is lower and decay much faster. The only reaction pathway with a positive TS barrier is the newly-predicted direct stripping pathway, shown in the right panel of Fig. 4, which reaction only occurs when $E_{\text{coll}} > 10$ kcal/mol, which is consistent with 10.38 kcal/mol TS barrier from *ab initio* calculation. Other reaction pathways do not have positive barriers.

38. The authors present a vast amount of quantum chemical data in this paper but do not include any of it in an appendix or supplementary information. To properly scrutinize this work, the geometries, energies, vibrational frequencies, T1 diagnostics, and other quantum chemical data should be provided to the reviewer and the audience. The statement that it is available by request is not sufficient.

Reply: Motivated by the Reviewer’s suggestion, we have provided the energies and Cartesian coordinates of all the stationary points in Supplementary Information.

39. The authors mention the new pathway to form $\text{CO}_2 + \text{H}_2 + \text{SO}_2$, but do not give a mechanism, energies, barriers, implications, etc. They say that an advantage of QCT is you get the geometries and can follow the reaction. If this is new and interesting, why did they not show a mechanism or delve deeper into the reaction? In the conclusion they write “This novel P3 channel promotes us to understand the mechanism in depth.” But they do not discuss the mechanism or if it is even possible.

Reply: We have added the following discussion for the new pathway to form $\text{CO}_2 + \text{H}_2 + \text{SO}_2$, on page 8, highlighted in red color:

“Following the trajectories forming P3, we see that as OCH_2O and SO_2 move away from each other, OCH_2O splits into two parts, i.e., CO_2 and H_2 . Finally, the product consists of three parts. We have not found any TS for this mechanism, thus it is probably a barrierless process.”

40. Some interesting points are discussed in the conclusions that would fit very nicely if placed in the discussion section and expanded upon. An example of which is “However, old channels present different behaviors: At low collision energies, reaction probabilities are high and they show flat behavior in the range of 0 to 8 bohr, later

show a declining trend and b_{\max} reaches largest values, i.e., 18 and 13 bohr at $E_{\text{coll}} = 1$ and 2 kcal/mol, respectively. It means this reaction can occur very easily, which agrees with the common idea that CH_2OO can react with SO_2 very fast.”

Reply: Following the Reviewer’s suggestion, we have expanded the discussion section, on page 12, highlighted in red color:

“The high and broad reaction probability curves at low collision energies show that the $\text{CH}_2\text{OO} + \text{SO}_2$ reaction is very fast at low temperature, which agrees with the common idea that CH_2OO can react with SO_2 rapidly.”

“The surprising relatively high reaction probabilities at high collision energies indicate that new complex mechanisms may exist for this reaction. However, we have not found new mechanisms at current stage, which needs additional work in the future.”

Referee reports: second round

REVIEWERS' COMMENTS:

Reviewer #2 (Remarks to the Author):

The revised manuscript is recommended to be published in CP

Reviewer #3 (Remarks to the Author):

Communications Chemistry - Reviewer Report

Manuscript COMMSCHEM-24-0034A

Title: Revealing new pathways for the reaction of Criegee intermediate CH_2OO with SO_2

This is a review of the above manuscript after first revision. I am pleased with the authors responses to my comments and the changes they implemented in the manuscript. I have learned from their responses, and I think this is a complete and well-done manuscript. The science was never in question, it was purely the presentation. The presentation of a potential new reaction mechanism is very interesting and hopefully the community will follow up with experiments confirming the pathway.

On final comments. This line that was added "which needs more experimental confirmation and atmospheric detection." I don't think it is possible to detect a mechanism in the atmosphere, so please remove that and finish with the "experimental confirmation".